# Severe Ground Fall Injury Associated with Alcohol Consumption in Geriatric Patients

**DOI:** 10.3390/healthcare10061111

**Published:** 2022-06-15

**Authors:** Jung Sung Hwang, Sun Hyu Kim

**Affiliations:** Department of Emergency Medicine, Ulsan University Hospital, University of Ulsan College of Medicine, 877 Bangeojinsunhwando-ro, Dong-gu, Ulsan 44033, Korea; 0735457@uuh.ulsan.kr

**Keywords:** emergency department-based injury in-depth surveillance, alcohol, falls, elderly

## Abstract

This study investigated characteristics of geriatric patients sustaining ground fall injury related to alcohol consumption and analyzed factors associated with the severity of such injuries in both alcohol- and non-alcohol-related cases. We retrospectively reviewed geriatric patients sustaining ground falls who were admitted to six university hospitals from 2011 to 2018. Patients were classified into alcohol and non-alcohol groups. Their general characteristics, clinical characteristics, and factors of injury severity were analyzed. A total of 31,177 patients were reviewed. Factors significantly associated with alcohol-related ground-fall injuries were: male, 65~84 years old, injury time other than 06:00~11:59, transportation to emergency department (ED) by ambulance and from other hospitals, injured in residential facilities, transportation areas, public or commercial facilities, activities other than paid or unpaid, non-slippery floor, obstacles, concrete floor, and absence of lights. Factors associated with severe injury after ground fall in alcohol-related cases were: male, injury time at between 06:00 and 17:59, transportation to the ED via ambulance from other hospitals, injured in residential facility, and slippery floor. Risk factors for severity in alcohol-related geriatric fall injury included: male sex, daytime injury, transportation by ambulance, injury in a residential facility, and slippery floor condition. Prophylactic measures and therapeutic interventions by ED teams are needed for the management of such cases.

## 1. Introduction

Currently, the geriatric population is experiencing rapid growth. According to the World Health Organization (WHO), the population of individuals older than 60 is expected to increase from 12% in 2015 to 22% in 2050 [1]. Life expectancy is also expected to increase due to the overall improvement in geriatric health [1]. Such an increase in the geriatric population worldwide has also led to an increase in traumatic injuries due to ground-level falls. This increase is an emerging health challenge [2,3,4].

Ground-level falls are the most common cause of injuries in the geriatric population [2,3]. They are sources of major trauma associated with morbidity and mortality [5,6]. Unintended falls are major causes of death in geriatric trauma [7]. The fall-related death rate is increasing, while death rates associated with other traumas are decreasing [2,8]. Characteristics of geriatric ground-level fall injury include higher admission rate, longer admission duration, and higher death rate compared to younger patients, although geriatric ground-level fall injury has a low-energy trauma mechanism [9,10] because of decrepitude in geriatric cases [5,6]. Several studies relevant to trauma mechanisms have been conducted. These studies have suggested that age alone might be a risk factor for severe trauma and worse outcomes [6,9,11]. In addition, alcohol consumption above the recommended limits is frequent in those of geriatric age [12,13,14]. Alcohol consumption can be a major risk factor for trauma and its severity in old age [14].

The demand for health care services for the geriatric population is increasing due to their decrepitude and decreased physiological reserve [5]. Furthermore, the cost of health care for older individuals is higher than that for younger age groups [1,9,10,15]. Alcohol consumption can especially deteriorate physical balance, movement, and response [2,16]. Therefore, identifying the role of multiple factors in alcohol consumption and other biological, behavioral, environmental, and socioeconomic variables [1,17] can reduce the health care burden by preventing ground-level falls in the geriatric population [1]. Although several studies have investigated the characteristics of geriatric fall-related injuries [1,2,4,5,7,11,14,18,19,20,21,22,23], studies investigating the characteristics of geriatric patients experiencing alcohol-related fall injuries and factors associated with major trauma have not been reported yet. Thus, the objective of this study was to analyze the characteristics of geriatric patients experiencing alcohol-related fall injuries and factors associated with major trauma.

## 2. Methods

### 2.1. Study Design

This retrospective case-control study investigated Emergency Department-based Injury In-depth Surveillance (EDIIS) data from the Korea Disease Control and Prevention Agency (KDCA) collected between 2011 and 2018. Patients were classified into alcohol-related and non-alcohol-related groups. Each group was then divided into two groups according to the severity of injuries: severe patients and non-severe patients.

### 2.2. Study Setting and Population

The patient group included patients aged 65 years or older who visited the emergency department of six university hospitals from January 2011 to December 2018 due to a ground-level fall.

### 2.3. Study Outcomes

The primary outcome was the correlation between geriatric ground-level fall injury and history of alcohol consumption. Secondary outcomes were factors related to severity in both alcohol-related and non-alcohol-related groups.

### 2.4. Data Sources and Measurement

Patients’ general characteristics, including sex, age, season of injury, time of injury, occupation, level of education, method of transportation to ED, location of injury, and activity during injury were examined. Characteristics of the floor during trauma injuries were examined. Based on age, patients were classified into three groups: 65~74 years, 75~84 years, and 85 years and older. The average age was also determined. Based on the season of ED admission (when patients were admitted to ED), patients were classified into four groups: Spring (March–May), Summer (June–August), Autumn (September–November), and Winter (December–February). Based on the time of injury, four groups were established at 6 h intervals: 00:00~05:59, 06:00~11:59, 12:00~17:59, and 18:00~23:59 h. The level of education was defined by the highest level achieved. Accordingly, patients were classified into four categories: uneducated or elementary school, junior high school, high school, and college. The mode of transportation to the ED was by public ambulance, transfer from another medical facility, or individual transportation. The location of injury occurrence was either residential facility, transportation area, public or commercial facility, or other areas. Injury might have occurred during paid or unpaid work (such as cooking, cleaning, and gardening), sports or leisure activity, daily activities (such as eating a meal or taking a shower), and other activities (such as religious duties and fights). The floor condition was classified as slippery or non-slippery. The type of floor was concrete, wooden, or earthen. The slope of the floor, the presence of obstacles, and lights were also examined.

Clinical characteristics of patients including results of ED treatment, severity, location of major injury, consciousness at ED, blood pressure at the time of ED arrival, and Excess Mortality Ratio-based Injury Severity Score (EMR-ISS) were investigated. Results of ED treatment were classified into five groups: discharge from ED, transfer to another facility, admission to general ward, admission to ICU, and death in ED. Severity was defined as hopeless discharge, transfer to other medical facilities due to lack of ICU or availability of emergency surgery (or further specialized treatment in a tertiary hospital), admission to ICU, need for emergency operation, and death in ED. Other patients were defined as the non-severe group. Based on the location of major injury, patients were classified into seven groups according to International Statistical Classification of Diseases and related Health Problems 10th Revision (ICD-10): head and neck (S00-19), thorax (S20~29), abdomen (S30~39), upper extremity (S40~69), hip and thigh (S70~79), lower extremity (S80~99), and multiple regions in the body (T00~T14). Based on consciousness in the ED, patients were classified into four groups according to the Alert, Verbal, Pain, Unresponsive (AVPU) scale. Systolic and diastolic pressure at the time of ED admission were also measured. EMR-ISS is generally utilized in South Korea as it is based on the ICD-10. EMRs for all ICD-10 codes from the Korean national injury database were used to grade the severity of every injury on a scale of 1~5. The EMR-ISS was calculated as the sum of the squares of three maximum severity grades [24].

### 2.5. Statistical Analysis

The chi-square test or Fisher’s exact test was used for categorical variables and Student’s *t*-test was used for numerical variables to compare general and clinical characteristics of alcohol-related and non-alcohol-related groups, or comparisons between severe and non-severe groups. Comparisons of proportions between different groups were performed using Pearson’s chi-square test and Fisher’s exact test. Normality of the distribution of all variables was tested with the Kolmogorov–Smirnov test. All variables were normally distributed. To reduce bias and potential confounding between two groups, adjustment was performed for different distributions of baseline characteristics (sex, time of injury, location of injury, activity during injury, age) using 2:1 propensity score matching analysis with the nearest neighbor method based on a greedy matching algorithm that could sort data by estimated propensity score. Propensity score matching was conducted by 2:1 nearest-neighbor matching using a caliper size of 0.2. Balance of covariates in the matched group was evaluated by measuring their standardized differences in means. In this analysis, standard mean difference was assumed to be the effect size. Univariate logistic regression analysis of patients’ general characteristics was performed to identify factors associated with alcohol-related ground-fall injury and severity of alcohol-related and unrelated groups. Multivariate logistic regression analysis (stepwise, forward) was performed by selecting statistically significant variables (*p*-value < 0.05) in the univariate logistic regression analysis. Education and occupation variables were not included in the logistic regression analysis due to a large number of missing values, since these parameters were evaluated only in admitted patients. IBM SPSS for windows version 24.0 (IBM Inc., Armonk, NY, USA) was used for statistical analyses and a *p*-value < 0.05 was defined as statistically significant.

## 3. Results

A total of 31,177 patients were enrolled. There were 2092 (6.7%) cases of alcohol-related patients and 29,085 (93.3%) non-alcohol-related patients. The alcohol-related group had a higher number of males (73.6%) than the non-alcohol-related group (35.9%) (Figure 1).

Regarding the age of subjects, the proportion of patients in the age range of 65–74 years was very high in the alcohol-related group (74.3%). Regarding the season of admission, Autumn was the most common season of injury in both groups. The most common time of injury in the alcohol-related group was 18:00~23:59 h (54.5%), higher than the time of 12:00~17:59 h (38.3%) in the non-alcohol-related group. The most common method of transportation to the ED was public ambulance in the alcohol-related group (69.1%) and individual transportation in the non-alcohol related group (44.6%). The most common location of injury occurrence was a transportation area in the alcohol-related group (63.0%) and residential facility in the non-alcohol-related group (59.0%). Among patients without alcohol issues, daily activity was the most frequent activity during injury (79.2%). Floor characteristics of individual groups were also investigated. In the alcohol-related group, the proportion of falls under non-slippery floor conditions (96.2%), sloping floors (15.9%), with presence of obstacles (12.2%), absence of lights (27.6%), and on concrete floors (98.5%) were higher than in the non-alcohol-related group. Results of ED treatment suggested that discharge was the most frequent in both groups. Patients were more frequently admitted to a general ward in the non-alcohol related group (33.2%), whereas patients were more frequently admitted to the ICU in the alcohol-related group (7.0%). Severe patients defined as ‘transfer to other facility, admitted patients, death at ED’ had a higher proportion in the alcohol-related group (9.0%). More patients with head and neck injuries (79.7%) and injuries to multiple body regions (7.3%) were found in the alcohol-related group, whereas more patients were non-alert at the ED in the alcohol-related group (11.0%) (Table 1).

The main factors significantly associated with alcohol-related ground-fall injury in elderly patients were: male sex, age ranges of 65 to 74 years and 75 to 84 years compared to age higher than 85 years, injury time other than 06:00~11:59 h, individual transportation and transportation to ED by public ambulance compared to from other medical facilities, injury occurring at a residential facility, transportation area, public or commercial facility compared to other area, sports or leisure activity, daily activity, or other activity compared to paid or unpaid work at the time of injury, non-slippery floor, presence of obstacles on the floor, absence of lights, and concrete floor (Table 2).

Additionally, 2:1 propensity score matching analysis was performed to adjust for different distributions of baseline characteristics (sex, time of injury, location of injury, activity during injury, age). All standardized mean differences in the baseline variables after 2:1 propensity score matching were <0.25 (25%), meaning that collinearity between baseline variables was evaluated adequately (Appendix A
Table A1 and Table A2).

Next, patients were divided into alcohol-related and non-alcohol-related groups to compare the severity of injuries. The proportion of men was higher in both groups. The most frequent time of injury was at 18:00~23:59 h. Public ambulance was the most frequent type of transportation to the ED in both groups. Transportation area was the most frequent location of injury. Among patients in the alcohol-related group, floor characteristics including slope (26.3%), presence of obstacles (18.8%), and darkness (33.9%) were more frequent in the severe group than in the non-severe group.

In the case of the non-alcohol related group, the proportion of men was higher in both groups. Autumn was the most frequent season of injury in both groups. Public ambulance was the most frequent method of transportation to the ED in the severe group (45.0%), whereas individual transportation was the most frequent method of transportation to the ED in the non-severe group (46.7%). The most frequent location of injury was residential facility in both groups. Daily activity was the most frequent time of injury in both groups of severity. Regarding floor characteristics, non-slippery (94.7%), sloping (13.9%), obstacles (11.9%), dark (20.1%), and other types of floors (7.1%) were more frequent in the severe group than in non-severe group (Table 3).

The main factors significantly associated with severe injury after ground fall in elderly patients with alcohol intake were: male sex, injury occurring at 06:00~11:59 h and 12:00~17:59 h compared to 18:00~23:59 h, transportation to ED by public ambulance and from other medical facilities compared to individual transportation, injury occurring at residential facility compared to transportation area, and slanted floor compared to flat floor (Table 4).

The main factors significantly associated with severe injury after ground fall in the elderly group with alcohol intake were: male sex, age between 65 and 74 years and 75 and 84 years compared to age ≥ 85 years, transportation to ED by public ambulance and other medical facilities compared to individual transportation, injury occurring at other areas and residential facilities compared to transportation area, paid or unpaid work or other activity compared to sports or leisure activity during injury, non-slippery floor, and presence of obstacles on floor. Absence of lights was significantly associated with non-severe injury after ground fall in non-alcohol ingested elderly patients (Table 5).

## 4. Discussion

This study compared characteristics of geriatric ground-level fall injuries and identified factors associated with alcohol intake. This study also identified factors associated with severity of injury in alcohol- and non-alcohol related groups. Male, relatively young age, time of injury other than morning, transportation by public ambulance or from other medical facility, injured in residential facility, transportation area, public or commercial facility, injured during other than paid or unpaid work, non-slippery floor, presence of obstacles, absence of lights, and concrete floor were factors associated with alcohol intake. Additionally, we conducted a 2:1 propensity score matching analysis to reduce the bias and potential confounding among patients in alcohol- and non-alcohol-related groups. When comparing before and after propensity score matching, age, season of injury, time of injury, and sloping floor were not valid. However, other variables were valid. Although ORs for variables were significantly decreased after propensity score matching, they still had valid significance. Factors associated with severity in alcohol-related groups were: male, injured at daytime, transportation by public ambulance or from other medical facilities, injured in a residential facility, and sloping floor. Meanwhile, factors associated with severity in the non-alcohol related group were: male, relatively young age, transportation by public ambulance or from other medical facilities, injured in a residential facility or other than transportation area, injured while not playing sports or enjoying a leisure activity, non-slippery floor with presence of obstacles, and absence of lights.

Fall injury is the main cause of trauma in all ages. It accounted for 10–15% of patients admitted to EDs [9,25]. Fall-related death is increasing, while death because of other mechanisms of trauma is stable or declining [1,8]. Prevention is the most efficient way to reduce traumatic events [9]. ED information is important when evaluating patients’ characteristics and establishing prevention strategies because the ED is the main avenue for the admission of geriatric patients with fall injuries [18,26].

Several studies conducted in the past have shown that men carry a higher risk of severe fall injury or fall-related mortality [2,6,9,10,11,22,27]. In this study, men manifested a higher risk of severe trauma in alcohol- and non-alcohol-related groups. Multiple factors play a role in fall injuries associated with older males, including increased outdoor activity, muscle weakness because of excessive physical labor, and underlying disease [2,10]. In addition, men display a cultural trend in that they do not generally visit a hospital unless they have severe illness [1,28]. Furthermore, more men drink alcohol than women.

In other studies, alcoholics were more likely to sustain severe injuries [1,18]. In this study, alcohol-ingesting patients were more likely to suffer from severe injuries and the proportion of unconsciousness was higher in these patients. Alcohol consumption weakens physical balance, movements, and responses. It also adversely affects self-defense response, which restricts the movements of arms during falls [2,16]. This can aggravate a fall injury.

Based on the time of occurrence, the proportion of alcohol-related injuries occurring during the nighttime was higher. Other studies investigating the relationship between time of admission to the ED and severity have reported unclear relationships [29,30]. In this study, the severity was higher when alcohol-consuming patients visited the ED during daytime. In spite of medical interventions, severity was higher during daytime because of the increased frequency of movements and activity. In contrast, alcohol-consuming patients showed fewer movements and activity at night.

Previous studies have investigated floor characteristics affecting the severity of fall injuries in geriatric populations [1,2,5,28,31,32] without considering the factor of alcohol consumption. In this study, the severity was higher in subjects consuming alcohol when they fell on a slanting floor. This might be due to poor physical balance [1,2,16].

This study showed that the characteristics of fall injuries and the characteristics of severe injuries in the elderly differed depending on their alcohol intake. For example, the high risk of severe injury on slanted floors rather than flat floors in a residential facility among elderly subjects ingesting alcohol is expected to influence the establishment of policies to improve the environment in order to prevent falls.

In our study, a total of 31,177 patients were included. Such a large sample size could weaken statistical significance. However, more than 2000 patients were assigned to both alcohol- and non-alcohol-related groups. In addition, the results of this study were not significantly different to those of previous studies [1,2,6,9,10,11,18,22,27]. Thus, sample size did not make a clinical difference by itself.

A major limitation in this study was the lack of individual medical histories and the use of medication, which could adversely affect fall injuries in geriatric subjects [2]. The severity of geriatric fall injuries varied with comorbidity, use of anticoagulant, and level of performance. For example, cardiovascular disease alone is associated with instability, postural dizziness, and syncope [7,33]. Antihypertensives, such as thiazide diuretics and beta blockers, can increase the risk of fall injuries [7,34]. Furthermore, cardiovascular diseases other than myocardial infarction and COPD can increase mortality due to fall-related injuries, which has been proven statistically [5]. The degree of cognitive disorder or independence in daily life, as well as visual impairment, can also affect the incidence rate and severity of geriatric fall injuries [2,23]. Therefore, additional studies that evaluate past medical history are needed. Furthermore, in the case of alcohol-ingesting patients, variables of ethanol level were not included, suggesting the need for studies to determine the correlation between ethanol level and severity. In addition, only patients treated at six university hospitals were included in this study, suggesting that injury severity could be relatively higher than in a population-based study. Moreover, severity criteria were classified according to the results of ED treatment. In the case of transfer to another hospital, such severity criteria might be subjective, suggesting the need for objective parameters, such as ISS, to evaluate severity.

In this study, previous evidence for ground-level injury was not changed. For example, alcohol ingestion and male sex increased the risk of more severe ground fall injuries, similar to other past studies [1,2,6,9,10,11,18,22,27]. However, factors associated with alcohol intake and severity in geriatric ground-level fall injuries were specified. Based on this study, we can establish plans to prevent geriatric ground-level fall injuries in both alcohol-related and non-alcohol-related groups and provide precautions for medical personnel in EDs who treat such patients. For example, daytime alcohol-ingested ground-level fall injury in residential facilities with a sloping floor suggests severe injury. Thus, protection measures, such as handrails should be installed in areas with sloping floors in residential facilities.

## 5. Conclusions

Factors related to increased risk of falls among elderly individuals consuming alcohol included: age between 65 and 84 years, injury at night, activities in transportation areas and public or commercial facilities, and during activity other than paid or unpaid work. Non-slippery floors, the presence of obstacles, concrete floors, and the absence of lights during injury increased the risk of fall injuries in the elderly ingesting alcohol. Therefore, prevention of falls among alcoholic elderly subjects living in such an environment requires the installation of safety devices or precautions.

The risk of severe ground-fall injury in elderly subjects who consume alcohol is increased by a later time of the injury during the day, occurrence in residential facility rather than transportation area, and occurrence on a slanted floor rather than flat floor. Therefore, medical personnel should adopt increased care during the management of alcoholics in the emergency department.

## Figures and Tables

**Figure 1 healthcare-10-01111-f001:**
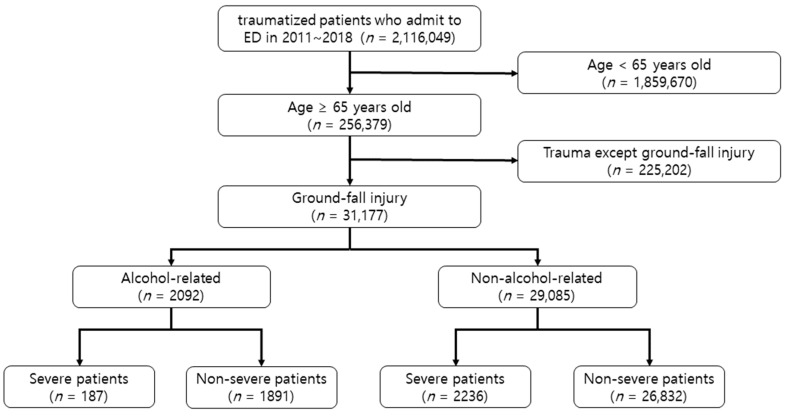
Flow chart of the study population. ED, Emergency room.

**Table 1 healthcare-10-01111-t001:** Characteristics of alcohol-related and unrelated ground-fall injuries in the elderly.

	Alcohol-Related	Non-Alcohol-Related	*p*-Value
	(N = 2092)	(N = 29,085)	
General characteristics			
Sex, male (%)	1540 (73.6)	10,434 (35.9)	<0.001
Age group, years old (%)			<0.001
65~74	1554 (74.3)	12,129 (41.7)	
75~84	762 (23.5)	12,256 (42.1)	
≥85	46 (2.2)	4700 (16.2)	
Average age (years)	71.4 ± 5.2	76.7 ± 7.4	<0.001
Season of injury (%)			0.008
Spring (March–May)	537 (25.7)	6552 (22.5)	
Summer (June–August)	478 (22.8)	6849 (23.5)	
Autumn (September–November)	551 (26.3)	8236 (28.3)	
Winter (December–February)	526 (25.1)	7448 (25.6)	
Time of injury (%)			<0.001
00:00–05:59 h	295 (14.1)	2063 (7.1)	
06:00–11:59 h	205 (9.8)	8064 (27.7)	
12:00–17:59 h	452 (21.6)	11,146 (38.3)	
18:00–23:59 h	1140 (54.5)	7812 (26.9)	
Occupation, employed (%)	N = 388	N = 11,792	0.811
	92 (23.7)	2732 (23.2)	
Education (%)	N = 144	N = 4356	0.001
Uneducated or elementary school	53 (36.8)	2369 (54.4)	
Junior high school	34 (23.6)	728 (16.7)	
High school	37 (25.7)	830 (19.1)	
≥College	20 (13.9)	429 (9.8)	
Transportation to ED (%)	N = 2087	N = 29,031	<0.001
Public ambulance	1443 (69.1)	11,482 (39.6)	
Other medical facility	142 (6.8)	4593 (15.8)	
Individual transportation	502 (24.1)	12,956 (44.6)	
Location of injury (%)			<0.001
Residential facility	430 (20.6)	17,151 (59.0)	
Transportation area	1318 (63.0)	6900 (23.7)	
Public or commercial facility	310 (14.8)	1816 (6.2)	
Other area	34 (1.6)	3218 (11.1)	
Activity during injury (%)	N = 2092	N = 29,083	<0.001
Paid or unpaid work	19 (0.9)	2437 (8.4)	
Sports or leisure activity	980 (46.8)	2893 (9.9)	
Daily activity	1069 (51.1)	23,026 (79.2)	
Others	24 (1.1)	727 (2.5)	
Slippery or non-slippery floor (%)	N = 2092	N = 29,079	<0.001
Slippery	80 (3.8)	2739 (9.4)	
Slope of floor, sloping or flat (%)	N = 2089	N = 29,068	<0.001
Sloping	333 (15.9)	3322 (11.4)	
Presence of obstacles, yes or none (%)	N = 2089	N = 29,072	<0.001
Yes	255 (12.2)	2807 (9.7)	
Presence of lights, none or yes (%)	N = 2089	N = 29,049	<0.001
None	576 (27.6)	5273 (18.2)	
Type of floor, concrete or others (%)	N = 2088	N = 28,982	<0.001
Concrete	2056 (98.5)	27,640 (95.4)	
Clinical characteristics			
Result of ED treatment (%)	N = 2078	N = 29,068	<0.001
Discharge	1679 (80.8)	16,995 (58.5)	
Transfer to other facility	47 (2.3)	780 (2.7)	
Admission to general ward	201 (9.7)	9655 (33.2)	
Admission to ICU	145 (7.0)	1497 (5.1)	
Death at ED	6 (0.3)	141 (0.5)	
Severe patients (%)	N = 2078	N = 29,068	0.033
	187 (9.0)	2236 (7.7)	
Major injury location (%)	N = 2082	N = 29,048	<0.001
Head and neck	1660 (79.7)	10,964 (37.7)	
Thorax	45 (2.2)	1735 (6.0)	
Abdomen	36 (1.7)	2581 (8.9)	
Upper extremity	116 (5.6)	3919 (13.5)	
Hip and thigh	40 (1.9)	6368 (21.9)	
Lower extremity	33 (1.6)	2058 (7.1)	
Multiple body regions	152 (7.3)	1423 (4.9)	
Consciousness at ED (%)	N = 1651	N = 22,619	<0.001
Alert	1470 (89.0)	21,812 (96.4)	
Verbal response	132 (8.0)	441 (1.9)	
Pain response	38 (2.3)	242 (1.1)	
Unresponsive	11 (0.7)	124 (0.5)	
Blood pressure (mmHg)	N = 1728	N = 23,993	<0.001
Systolic blood pressure	130.3 ± 24.4	143.4 ± 27.5	<0.001
Diastolic blood pressure	76.0 ± 14.4	79.2 ± 15.0	
EMR-ISS	N = 2087	N = 29,033	<0.001
	15.4 ± 12.2	14.0 ± 10.3	

ED, emergency department, ICU, intensive care unit, EMR-ISS, Excess Mortality Ratio-adjusted Injury Severity Score.

**Table 2 healthcare-10-01111-t002:** Factors associated with alcohol intake and ground-fall injuries in the elderly.

Univariate				Multivariate			
	OR	95% C.I.	*p*-Value		OR	95% C.I.	*p*-Value
Male sex vs. female	4.99	4.51–5.51	<0.001	Male sex vs. female	4.07	3.63–4.57	<0.001
Age group, years old				Age group, years old			
≥85	1.00			≥85	1.00		
65–74 years	13.09	9.75–17.59	<0.001	65–74 years	8.31	6.10–11.32	<0.001
75–84 years	4.10	3.03–5.56	<0.001	75–84 years	3.14	2.28–4.31	<0.001
Season of injury							
Autumn	1.00						
Winter	1.06	0.93–1.19	0.390				
Spring	1.23	1.08–1.39	0.001				
Summer	1.04	0.92–1.18	0.513				
Time of injury				Time of injury			
06:00–11:59 h	1.00			06:00–11:59 h	1.00		
12:00–17:59 h	1.60	1.35–1.89	<0.001	12:00–17:59 h	1.41	1.18–1.70	<0.001
18:00–23:59 h	5.74	4.93–6.68	<0.001	18:00–23:59 h	4.22	3.56–5.00	<0.001
00:00–05:59 h	5.63	4.68–6.77	<0.001	00:00–05:59 h	4.76	3.84–5.91	<0.001
Transportation to ED				Transportation to ED			
Other medical facility	1.00			Other medical facility	1.00		
Public ambulance	4.07	3.41–4.85	<0.001	Public ambulance	3.89	3.44–4.40	<0.001
Individual transportation	1.25	1.04–1.51	0.019	Individual transportation	1.24	1.00–1.54	0.048
Location of injury				Location of injury			
Other area	1.00			Other area	1.00		
Residential facility	2.37	1.67–3.37	<0.001	Residential facility	3.91	2.62–5.84	<0.001
Transportation area	18.08	12.83–25.48	<0.001	Transportation area	16.15	10.93–23.85	<0.001
Public or commercial facility	16.16	11.29–23.13	<0.001	Public or commercial facility	12.15	8.10–18.24	<0.001
Activity during injury				Activity during injury			
Paid or unpaid work	1.00			Paid or unpaid work	1.00		
Sports or leisure activity	43.45	27.51–68.63	<0.001	Sports or leisure activity	33.09	20.54–53.29	<0.001
Daily activity	5.96	3.78–9.39	<0.001	Daily activity	6.17	3.85–9.89	<0.001
Others	4.23	2.31–7.77	<0.001	Others	6.28	3.27–12.04	<0.001
Non-slippery condition of floor vs. slippery	2.62	2.08–3.28	<0.001	Non-slippery condition of floor vs. slippery	3.14	2.42–4.08	<0.001
Sloping vs. flat floor	1.47	1.30–1.66	<0.001				
Presence of obstacles on floor	1.30	1.14–1.49	<0.001	Presence of obstacles on floor	1.33	1.13–1.58	0.001
Absence of lights	1.72	1.55–1.90	<0.001	Absence of lights	2.13	1.87–2.42	<0.001
Concrete floor vs. other types	3.12	2.19–4.44	<0.001	Concrete floor vs. other types	1.92	1.25–2.94	<0.001

ED, emergency department.

**Table 3 healthcare-10-01111-t003:** General characteristics of alcohol- and non-alcohol-related ground-fall injuries in the elderly according to severity of injury.

Alcohol-Related				Non-Alcohol-Related			
	Severe (N = 187)	Non-Severe (N = 1891)	*p*-Value		Severe (N = 2236)	Non-Severe (N = 26,832)	*p*-Value
Sex, male (%)	154 (82.4)	1378 (72.9)	0.005	Sex, male (%)	1090 (48.7)	9334 (34.8)	<0.001
Age group, years old (%)			0.775	Age group, years old (%)			0.053
65–74 years	141 (75.4)	1399 (74.0)		65–74 years	964 (43.1)	11,155 (41.6)	
75–84 years	41 (21.9)	451 (23.8)		75–84 years	950 (42.5)	11,301 (42.1)	
≥85	5 (2.7)	41 (2.2)		≥85	322 (14.4)	4376 (16.3)	
Average age (years)	71.2 ± 5.1	17.4 ± 5.2	0.527	Average age (years)	76.4 ± 7.2	76.7 ± 7.4	0.019
Season of injury (%)			0.401	Season of injury (%)			0.045
Spring	49 (26.2)	486 (25.7)		Spring	503 (22.5)	6041 (22.5)	
Summer	51 (27.3)	422 (22.3)		Summer	524 (2.4)	6321 (23.6)	
Autumn	43 (23.0)	504 (26.7)		Autumn	682 (30.5)	7551 (28.1)	
Winter	44 (23.5)	479 (25.3)		Winter	527 (23.6)	6919 (25.8)	
Time of injury (%)			0.001	Time of injury (%)			0.165
00:00–05:59 h	26 (13.9)	265 (14.0)		00:00–05:59 h	171 (7.6)	1889 (7.0)	
06:00–11:59 h	31 (16.6)	174 (9.2)		06:00–11:59 h	579 (25.9)	7482 (27.9)	
12:00–17:59 h	49 (26.2)	401 (21.2)		12:00–17:59 h	886 (39.6)	10,253 (38.2)	
18:00–23:59 h	81 (43.3)	1051 (55.6)		18:00–23:59 h	600 (26.8)	7208 (26.9)	
Occupation, employed (%)	N = 16151 (31.7)	N = 22741 (18.1)	0.002	Occupation, employed (%)	N = 1934519 (26.8)	N = 98572213 (22.5)	<0.001
Education (%)	N = 68	N = 76	0.687	Education (%)	N = 740	N = 3616	0.015
Uneducated or elementary school	23 (33.8)	30 (39.5)		Uneducated or elementary school	363 (49.1)	2006 (55.5)	
Junior high school	18 (26.5)	16 (21.1)		Junior high school	141 (19.1)	587 (16.2)	
High school	16 (23.5)	21 (27.6)		High school	153 (20.7)	677 (18.7)	
≥college	11 (16.2)	9 (11.8)		≥college	83 (11.2)	346 (9.6)	
Transportation to ED (%)	N = 187	N = 1886	<0.001	Transportation to ED (%)	N = 2233	N = 26,781	<0.001
Public ambulance	109 (58.3)	1323 (70.1)		Public ambulance	1004 (45.0)	10,467 (39.1)	
Other medical facility	54 (28.9)	88 (4.7)		Other medical facility	792 (35.5)	3800 (14.2)	
Individual transportation	24 (12.8)	475 (25.2)		Individual transportation	437 (19.6)	12,514 (46.7)	
Location of injury (%)			<0.001	Location of injury (%)			<0.001
Residential facility	62 (33.2)	367 (19.4)		Residential facility	1318 (58.9)	15,830 (59.0)	
Transportation area	84 (44.9)	1225 (64.8)		Transportation area	395 (17.7)	6495 (24.2)	
Public or commercial facility	35 (18.7)	272 (14.4)		Public or commercial facility	125 (5.6)	1690 (6.3)	
Other area	6 (3.2)	27 (1.4)		Other area	398 (17.8)	2817 (10.5)	
Activity during injury (%)			0.149	Activity during injury (%)	N = 2236	N = 26,830	<0.001
Paid or unpaid work	3 (1.6)	16 (0.8)		Paid or unpaid work	344 (15.4)	2093 (7.8)	
Sports or leisure activity	84 (44.9)	888 (47.0)		Sports or leisure activity	173 (7.7)	2717 (10.1)	
Daily activity	95 (50.8)	968 (51.2)		Daily activity	1580 (70.7)	21,432 (79.9)	
Others	5 (2.7)	19 (1.0)		Others	139 (6.2)	588 (2.2)	
Slippery or non-slippery floor (%)			0.110	Slippery or non-slippery floor (%)			<0.001
Slippery	3 (1.6)	76 (4.0)		Slippery	119 (5.3)	2618 (9.8)	
Slope of floor, sloping or flat (%)	N = 186	N = 1889	<0.001	Slope of floor, sloping or flat (%)	N = 2233	N = 26,818	<0.001
Sloping	49 (26.3)	282 (14.9)		Sloping	310 (13.9)	3010 (11.2)	
Presence of obstacles, yes or none (%)	N = 186	N = 1889	0.007	Presence of obstacles, yes or none (%)	N = 2233	N = 26,822	<0.001
Yes	35 (18.8)	219 (11.6)		Yes	265 (11.9)	2540 (9.5)	
Presence of lights, none or yes (%)	N = 186	N = 1889	0.048	Presence of lights, none or yes (%)	N = 2233	N = 26,799	0.016
None	63 (33.9)	509 (26.9)		None	448 (20.1)	4822 (18.0)	
Type of floor, concrete or others (%)	N = 186	N = 1888	0.061	Type of floor, concrete or others (%)	N = 2220	N = 26,745	<0.001
Concrete	180 (96.8)	1862 (98.6)		Concrete	2062 (92.9)	25,562 (95.6)	

ED, emergency department.

**Table 4 healthcare-10-01111-t004:** Factors associated with severe injury after ground fall in elderly subjects ingesting alcohol.

Univariate				Multivariate			
	OR	95% C.I.	*p*-Value		OR	95% C.I.	*p*-Value
Male sex vs. female	1.74	1.18–2.56	0.005	Male sex vs. female	1.78	1.18–2.70	0.007
Age group, years old							
75-84	1.00						
65-74	1.11	0.77–1.60	0.578				
≥85	1.34	0.50–3.59	0.558				
Season of injury							
Autumn	1.00						
Winter	1.08	0.69–1.67	0.741				
Spring	1.18	0.77–1.81	0.445				
Summer	1.42	0.93–2.17	0.109				
Time of injury				Time of injury			
18:00–23:59 h	1.00			18:00–23:59 h	1.00		
00:00–05:59 h	1.27	0.80–2.02	0.306				
06:00–11:59 h	2.31	1.48–3.60	<0.001	06:00–11:59 h	1.66	1.02–2.71	0.043
12:00–17:59 h	1.59	1.09–2.30	0.015	12:00–17:59 h	1.58	1.06–2.34	0.024
Transportation to ED				Transportation to ED			
Individual transportation	1.00			Individual transportation	1.00		
Public ambulance	1.63	1.04–2.57	0.035	Public ambulance	1.69	1.06–2.70	0.028
Other medical facility	12.15	7.14–20.67	<0.001	Other medical facility	12.69	7.30–22.08	<0.001
Location of injury				Location of injury			
Transportation area	1.00			Transportation area	1.00		
Public or commercial facility	1.88	1.24–2.84	0.003				
Other area	3.24	1.30–8.07	0.011				
Residential facility	2.46	1.74–3.49	<0.001	Residential facility	2.26	1.55–3.30	<0.001
Activity during injury							
Sports or leisure activity	1.00						
Paid or unpaid work	1.98	0.57–6.94	0.285				
Daily activity	1.04	0.76–1.41	0.814				
Others	2.78	1.01–7.64	0.047				
Non-slippery condition of floor vs. slippery	2.57	0.80–8.22	0.112				
Sloping vs. flat floor	2.04	1.44–2.89	<0.001	Sloping vs. flat floor	1.83	1.22–2.75	0.004
Presence of obstacles on floor	1.77	1.19–2.62	0.005				
Absence of lights	1.39	1.01–1.91	0.044				
Other types vs. concrete	2.39	0.97–5.88	0.058				

ED, emergency department.

**Table 5 healthcare-10-01111-t005:** Factors associated with severe injury after ground fall in non-alcoholic elderly subjects.

Univariate				Multivariate			
	OR	95% C.I.	*p*-Value		OR	95% C.I.	*p*-Value
Male sex vs. female	1.78	1.64–1.94	<0.001	Male sex vs. female	1.67	1.52–1.83	<0.001
Age group, years old				Age group, years old			
≥85	1.00			≥85	1.00		
75–84 years	1.17	1.03–1.34	0.016	75–84 years	1.33	1.16–1.53	<0.001
65–74 years	1.14	1.00–1.30	0.047	65–74 years	1.20	1.05–1.37	0.008
Season of injury							
Winter	1.00						
Spring	1.09	0.96–1.24	0.169				
Summer	1.09	0.96–1.23	0.186				
Autumn	1.19	1.05–1.34	0.005				
Time of injury							
06:00–11:59 h	1.00						
00:00–05:59 h	1.17	0.98–1.40	0.084				
12:00–17:59 h	1.12	1.00–1.25	0.047				
18:00–23:59 h	1.08	0.96–1.21	0.228				
Transportation to ED				Transportation to ED			
Individual transportation	1.00			Individual transportation	1.00		
Public ambulance	2.75	2.45–3.08	<0.001	Public ambulance	2.77	2.47–3.12	<0.001
Other medical facility	5.97	5.28–6.75	<0.001	Other medical facility	5.83	5.14–6.62	<0.001
Location of injury				Location of injury			
Transportation area	1.00			Transportation area	1.00		
Public or commercial facility	1.22	0.99–1.50	0.065				
Other area	2.32	2.01–2.69	<0.001	Other area	1.27	1.07–1.50	0.005
Residential facility	1.37	1.2–1.54	<0.001	Residential facility	1.24	1.10–1.41	<0.001
Activity during injury				Activity during injury			
Sports or leisure activity	1.00			Sports or leisure activity	1.00		
Daily activity	1.16	0.99–1.36	0.076				
Others	3.71	2.92–4.72	<0.001	Others	2.80	2.17–3.62	<0.001
Paid or unpaid work	2.58	2.13–3.13	<0.001	Paid or unpaid work	1.84	1.50–2.26	<0.001
Non-slippery condition of floor vs. slippery	1.92	1.59–2.32	<0.001	Non-slippery condition of floor vs. slippery	1.64	1.36–1.99	<0.001
Sloping floor vs. flat	1.28	1.12–1.45	<0.001				
Presence of obstacles on floor	1.29	1.13–1.47	<0.001	Presence of obstacles on floor	1.40	1.21–1.61	<0.001
Absence of lights	1.14	1.03–1.28	0.015	Absence of lights	0.86	0.76–0.96	0.010
Other types vs. concrete	1.66	1.39–1.97	<0.001				

ED, emergency department.

## Data Availability

The Korea Disease Control and prevention Agency (KDCA) has the authority to access and use these data. Data of this study can be requested through the Department of Injury Prevention of the KDCA by E-mail (kcdcinjury@korea.kr) or website (http://www.kdca.go.kr/, accessed on 28 January 2022).

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
