# Peer review of "Severe Ground Fall Injury Associated with Alcohol Consumption in Geriatric Patients"

_healthcare, 2022, doi:10.3390/healthcare10061111_

Round 1
Reviewer 1 Report
General: edit based on academic scientific writing.
Method: add STROBE criteria checklist.
Method: add patient flow figure
patients were classified into 3 groups: add the reason with citation.
Discussion: 1st paragraph: write summary of resuls
Discussion: write clinical and research implication clearly.
your results overcome previous evidence or clinical practice?
Discussion: As you pointed out, your results show statistically difference due to large sample size. There also means clinical difference? Please discuss it.
Author Response
Attached file is the answers for comment's of reviewer 1.

Reviewer 2 Report
The authors presented interesting results on.
- In line 63, what is the season here? season of the admission?
- Please provide a flowchart on the inclusion and exclusion of the study population?
- In all tables, if p-value is less than 0.001, please use p<0.001 instead of 0.000.
- Given the imbalanced data in the outcome variable in analyses shown in Table 2 and 3, have authors considered weighted logistic regressions instead?
- In the multivariate logistic regression analysis, it might not suffice to fit with variables with p-value < 0.05. Please consider stepwise, forward, or backward regression to obtain the best fit. In addition, no evaluation of collinearity between variables are performed.
I will continue reviewing the rest of the paper once the comments on statistical analyses are addressed and updated.
Author Response
Attached file is the answers for comment's of reviewer 2.

Round 2
Reviewer 1 Report
Thank you for revision.
Describe your suggestion on Q6 and Q7 of author response in discussion briefly. I think it will be useful as research and clinical implication.
Author Response
Reviewer 1
Q1. Describe your suggestion on Q6 and Q7 of author response in discussion briefly. I think it will be useful as research and clinical implication.
- A) We thank the reviewer for pointing this out and we agree with the reviewer that we need to discuss this. Therefore, we have added our suggestion on Q6 and Q7 in the discussion section briefly as shown below:
Added
In our study, a total of 31,177 patients were included. Such large sample size could weaken the statistical significance. However, more than 2,000 patients were assigned to each of alcohol and non-alcohol related groups. In addition, results of this study were not significantly different from those of previous studies [1-2, 6, 9-11, 18, 22, 27]. Thus, sample size did not make a clinical difference by itself.
In this study, previous evidence for ground-level injury was not changed. For example, alcohol ingestion and male sex increased the risk for the severity of ground fall injury, similar to other past studies [1-2, 6, 9-11, 18, 22, 27]. However, factors associated with alcohol-relation and severity in geriatric ground-level fall injuries were specified.

Reviewer 2 Report
Thank you so much for addressing all my comments. Here are a few more follow-up comments.
1. In Table 1, there are still p values of 0.000. Again, please change them to "<0.001".
2. In the flow chart, please use commas to separate groups of thousands and a period as a decimal separator, like, 2,116,049.
3. Please consider adding a line divider to separate variables in all tables as it is hard to tell which rows belong to the same variable.
4. There is no further discussion on comparing results between Table 2 and appendix 2.
